# Enhancing the Medical Context-Awareness Ability of LLMs via Multifaceted Self-Refinement Learning

## Abstract

Large language models (LLMs) have shown great promise in the medical domain, achieving strong performance on several benchmarks. However, they continue to underperform in real-world medical scenarios, which often demand stronger context-awareness, i.e., the ability to recognize missing or critical details (e.g., user identity, medical history, risk factors) and provide safe, helpful, and contextually appropriate responses. To address this issue, we propose Multifaceted Self-Refinement (MuSeR), a data-driven approach that enhances LLMs' context-awareness along three key facets (decision-making, communication, and safety) through self-evaluation and refinement. Specifically, we first design a attribute-conditioned query generator that simulates diverse real-world user contexts by varying attributes such as role, geographic region, intent, and degree of information ambiguity. An LLM then responds to these queries, self-evaluates its answers along three key facets, and refines its responses to better align with the requirements of each facet. Finally, the queries and refined responses are used for supervised fine-tuning to reinforce the model's context-awareness ability. Evaluation results on the latest HealthBench dataset demonstrate that our method significantly improves LLM performance across multiple aspects, with particularly notable gains in the context-awareness axis. Furthermore, by incorporating knowledge distillation with the proposed method, the performance of a smaller backbone LLM (e.g., Qwen3-32B) surpasses its teacher model, achieving a new SOTA across all open-source LLMs on HealthBench (63.8%) and its hard subset (43.1%). Code and dataset will be released at `https://anonymous.4open.science/r/MuSeR-EC43`.

## 1 Introduction

Large language models (LLMs) have witnessed significant advancements in recent years (Achiam et al., 2023; Anil et al., 2023; Dubey et al., 2024; Liu et al., 2024; Yang et al., 2025) and demonstrated promising capabilities in various domains, including the medical domain. Recent studies (Singhal et al., 2023a;b; Nori et al., 2023a; Qiu et al., 2024; Liu et al., 2025) indicate that current LLMs (e.g., GPT-4) encode substantial medical knowledge and achieves strong performance on several medical benchmarks. Despite these advancements, LLMs still struggle to meet the demands of real-world medical applications, limiting their practical utility in healthcare settings.

One of the decisive differences between medical benchmark questions and real-world scenarios lies in the requirement for stronger *context-awareness*, namely, the ability to recognize missing or critical details (e.g., medical history, user identity, risk factors) and to provide safe, helpful, and contextually appropriate responses. As illustrated in Figure 1a, existing medical benchmarks typically adopt question-answering task for evaluation, where questions contain sufficient information for answering, presented in the impersonal tone, and answering errors have limited consequences. In contrast, real-world medical scenarios often omit key details for decision-making, involve diverse user roles (e.g., patients, doctors), and require careful consideration of safety and ethical implications. While current LLMs perform well on exam-style context, they often overlook the contextual factors in real-world medical scenarios, leading to responses that may be inappropriate, unsafe, or unhelpful to the user's specific situation.

Figure 1: (a) Comparison between medical exam questions and real-world medical scenarios. (b) The proposed Multifaceted Self-Refinement learning framework (**MuSeR**) to enhance the medical context-awareness ability of LLMs through data synthesis and self-refinement.

In this paper, we aim to enhance the context-awareness of LLMs in the medical domain. A common approach is to collect high-quality real-world medical conversations for supervised fine-tuning (SFT). However, this is often impractical due to high collection costs and ethical concerns. To address this, we explore a cost-effective and scalable alternative: enhancing context-awareness through data synthesis. Specifically, we propose a novel **Mu**ltifaceted **Se**lf-**R**efinement (**MuSeR**) framework. MuSeR improves medical context-awareness by synthesizing simulated real-world medical queries and generating context-aware responses by self-refining the answers of LLMs along three key facets of context-awareness: decision-making, communication, and safety. As shown in Figure 1b, the proposed framework consists of three main components: (1) a attribute-conditioned query generator that simulates diverse real-world user contexts by varying attributes such as role, geographic region, intent, and degree of information ambiguity; (2) a multifaceted self-refinement module where an LLM responds to the generated queries, evaluates its answers along the three key facets, and refines its responses to better align with the requirements of each facet; and (3) a supervised fine-tuning stage where the generated queries and refined responses are used to reinforce the model's context-awareness ability. The entire process does not require any external medical corpora or human annotations, making it a cost-effective and scalable solution for enhancing LLMs' context-awareness in the medical domain.

To evaluate the effectiveness of our proposed method, we apply the proposed method on different sizes of LLMs (Qwen3-32B, Qwen3-14B, OpenPangu-7B) and assess their performance on the latest HealthBench dataset (Arora et al., 2025), which focuses on evaluating LLMs performance in real-world medical scenarios. The results demonstrate that our method significantly improves LLM performance on HealthBench, with particularly notable gains in the context-awareness axis. Furthermore, by incorporating knowledge distillation into the proposed framework using a strong teacher model (e.g., GPT-oss-120B), the performance of a smaller backbone LLM (e.g., Qwen3-32B) surpasses that of the teacher model by 6%, achieving a new state-of-the-art result among open-source LLMs on HealthBench (**63.8**%) and its hard subset (**43.1**%). Our main contributions are summarized as follows:

- We propose a novel Multifaceted Self-Refinement (**MuSeR**) learning framework that enhances LLMs' context-awareness across three key facets (decision-making, communication, and safety) through self-evaluation and refinement, facilitating their application in real-world medical scenarios.

- Extensive experiments on the HealthBench dataset demonstrate the effectiveness of our method in improving LLM performance, particularly in the context-awareness axis.

- By incorporating knowledge distillation into our framework, we achieve new state-of-the-art performance among open-source LLMs on the HealthBench dataset (**63.8**%) and the hard subset (**43.1**%) using only one million generated queries.

## 2 RELATED WORK

**LLM Medical Evaluation**   Most of existing medical benchmarks for LLMs are in the question-answering form, where the questions are sourced from medical exams (Vilares & Gómez-Rodríguez, 2019; Jin et al., 2021; Pal et al., 2022; Cai et al., 2024; Wang et al., 2024; Qiu et al., 2024; Zhou et al., 2024), literatures (Jin et al., 2019; Krithara et al., 2023), and healthcare consultations (Liu et al., 2020; Abacha et al., 2021; Singhal et al., 2023a). These benchmarks primarily evaluate the LLMs' medical knowledge and reasoning abilities, and existing LLMs are reported to achieve strong performance on these benchmarks (Singhal et al., 2023a;b; Nori et al., 2023b; Qiu et al., 2024). For example, GPT-4 achieves over 90% accuracy on the MedQA-USMLE exam dataset, approaching the performance level of human medical experts. Nevertheless, these benchmarks may not fully capture the complexities of real-world medical scenarios, especially in terms of context-awareness. Recently, OpenAI proposed **HealthBench** (Arora et al., 2025), a new benchmark includes 5,000 realistic health conversations annotated by 262 physicians across 60 countries, evaluating LLMs' performance as medical assistants in real-world scenarios. In this work, we primarily evaluate the effectiveness of our method on the HealthBench dataset.

**Medical LLM Training**   Existing works on training medical LLMs mainly focus on two aspects: (1) continual pre-training on medical corpora to inject domain-specific knowledge into LLMs (Chen et al., 2023; Qiu et al., 2024; Zhang et al., 2024); (2) post training on downstream tasks to enhance the model's reasoning and decision-making capabilities (Singhal et al., 2023a;b; Toma et al., 2023; Christophe et al., 2024b;a; Chen et al., 2025b). For the training data, most works utilize existing medical corpora, such as PubMed articles (Roberts, 2001), clinical notes (Johnson et al., 2020; Zhao et al., 2023), and medical QA datasets (Jin et al., 2021; Pal et al., 2022). Due to the data availability and privacy concerns, recent works (Bai et al., 2024; Das et al., 2024; Corbeil et al., 2025) start to leverage LLM-generated synthetic data for training medical LLMs and show promising results. While these methods effectively improve LLMs' medical knowledge mastery and reasoning skills, our work mainly focuses on enhancing the context-awareness ability of LLMs, which is also a crucial aspect for LLMs' practical application in the medical domain.

**Knowledge Distillation**   Knowledge distillation (KD) (Hinton et al., 2015; Sanh et al., 2019; Jiao et al., 2019) transfers knowledge from a large teacher model to a smaller student model, enabling efficiency while maintaining performance. In the LLM era, knowledge distillation is typically performed by generating distillation data using a strong teacher LLM for fine-tuning the student LLM in both general domain (Abdin et al., 2024; Yang et al., 2025; Guo et al., 2025) and medical domain (Zhang et al., 2023; Chen et al.). In this work, we explore the value of our framework in knowledge distillation and demonstrate that our method is effective in generating high-quality queries for knowledge distillation.

## 3 METHODOLOGY

In this section, we present our proposed Multifaceted Self-Refinement (MuSeR) learning framework to enhance the context-awareness ability of LLMs in the medical domain. An overview of the proposed framework is illustrated in Figure 2. In the following sections, we first formulate the problem and then detail the design of each component in the framework.

### 3.1 PROBLEM FORMULATION

Our goal is to improve the medical context-awareness of an LLM $\mathcal{M}$, such that it provides safe, helpful, and contextually appropriate responses to real-world medical queries. Let $q \sim P^*(\cdot)$ denote a real-world medical query where $P^*(\cdot)$ is the distribution of real-world medical queries, $P_{\mathcal{M}}(\cdot|q)$ denote the model's conditional response distribution, and $P^*(\cdot|q)$ denote the ideal conditional response distribution, where a response $r \sim P^*(\cdot|q)$ attends to the contextual information of $q$ across a set of facets $f_1, f_2, \cdots, f_N$. Conceptually, our goal can be expressed as reducing the divergence between these conditional distributions across queries:

$$\mathcal{M}^* = \arg\min_{\mathcal{M}} \mathbb{E}_{q \sim P^*(\cdot)} \left[ \text{KL} \left( P^*(\cdot|q) || P_{\mathcal{M}}(\cdot|q) \right) \right].  \quad (1)$$

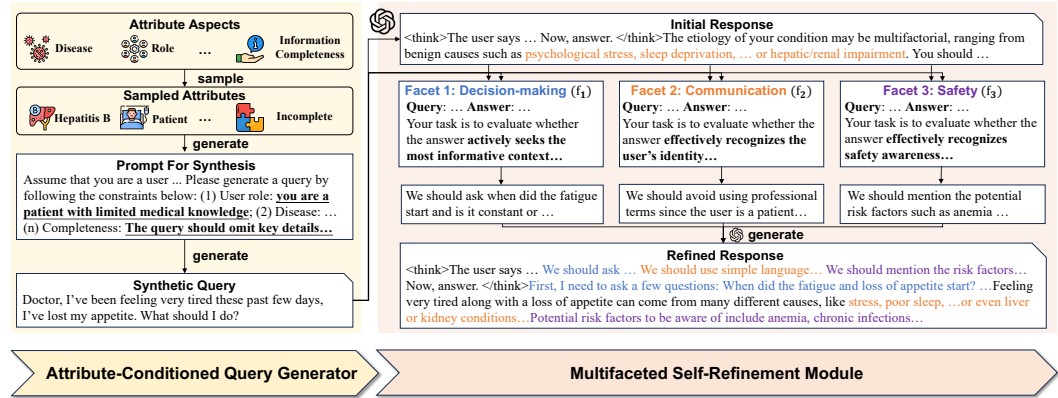

Figure 2: An overview of the proposed Multifaceted Self-Refinement (**MuSeR**) learning framework, with the SFT-based context-awareness enhancement stage omitted for simplicity.

where $\mathrm{KL}(\cdot||\cdot)$ denotes the KL divergence. However, the real-world query distribution $P^*(\cdot)$ and response distribution $P^*(\cdot|q)$ are typically inaccessible in practical scenarios. To address this, we aim to (1) construct a query generator G that induces a distribution $P_\mathrm{G}(\cdot)$ to approximate $P^*(\cdot)$:

$$\mathrm{G} \approx \arg\min_{\mathrm{G}'} \mathrm{KL}\left(P^*(\cdot)||P_{\mathrm{G}'}(\cdot)\right), \tag{2}$$

(2) develop a response generator R that induces a distribution $P_\mathrm{R}(\cdot|q)$ to approximate $P^*(\cdot|q)$ for any $q \sim P_G$:

$$\mathrm{R} \approx \arg\min_{\mathrm{R}'} \mathbb{E}_{q\sim P_\mathrm{G}(\cdot)}\left[\mathrm{KL}\left(P^*(\cdot|q)||P_{\mathrm{R}'}(\cdot|q)\right)\right], \tag{3}$$

(3) optimize the model $\mathcal{M}$ such that its response distribution $P_\mathcal{M}(\cdot|q)$ is aligned with $P_\mathrm{R}(\cdot|q)$:

$$\mathcal{M}^* \approx \arg\min_{\mathcal{M}'} \mathbb{E}_{q\sim P_\mathrm{G}(\cdot)}\left[\mathrm{KL}\left(P_\mathrm{R}(\cdot|q)||P_{\mathcal{M}'}(\cdot|q)\right)\right]. \tag{4}$$

Note that the formulations above represent our design goals rather than explicit optimization objectives. In the following sections, we describe the proposed learning framework in detail, including the facets of context-awareness it incorporates, the design of the query generator G and response generator R, and the training strategy for optimizing the model $\mathcal{M}$.

### 3.2 Multifaceted Self-Refinement Learning Framework

**Facets of Context-Awareness (f)**  We primarily consider three key facets of context-awareness that are crucial for providing safe, helpful, and appropriate responses in the medical domain:

- **Decision-Making Awareness** ($f_1$): This facet focuses on identifying critical information (e.g., medical history, medication, examination results) essential for accurate medical decision-making, as well as actively seeking missing details from users when necessary. Such awareness is critical for ensuring the accuracy and practical utility of medical advice.

- **Communication Awareness** ($f_2$): This facet involves recognizing the user's identity (e.g., patient, doctor) and response preferences, and tailoring both terminology (e.g., layman vs. professional) and level of detail (e.g., brief vs. comprehensive) accordingly. This facet is essential for providing responses that match the user's knowledge background and expectations.

- **Safety Awareness** ($f_3$): This facet requires the model to recognize potential risk factors (e.g., symptom severity, underlying conditions) and ethical considerations (e.g., the use of unproven drugs) in its responses. Such awareness is vital for ensuring both the safety and ethical integrity of the medical advice provided.

**Attribute-Conditioned Query Generation** ($G$)  For the query generator $G(\cdot)$, to simulate the complexity of real-world query distribution $P^*(\cdot)$, we assume that the real-world query is controlled by a set of attributes $\mathbf{a} = \{a_1, a_2, \cdots, a_N\}$ (e.g., user role, intent), such that $P_{real}(q) =$

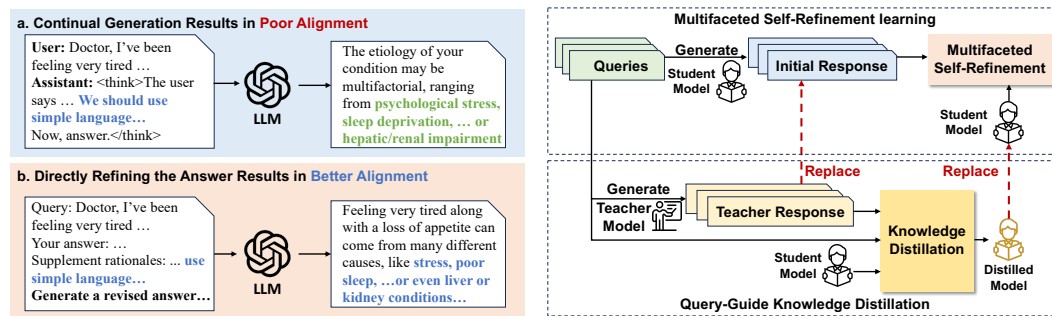

Figure 3: Comparison between two strategies for answer generation: (1) continual generation conditioned on the refined reasoning, and (2) answer refinement guided by multifaceted rationales.

Figure 4: Query-Guided Knowledge Distillation integrated with the Multifaceted Self-Refinement (MuSeR) learning framework for enhancing medical context-awareness.

$P(q|\mathbf{a})P(\mathbf{a})$. Built on that, the proposed attribute-conditioned query generator first samples a set of attributes $\mathbf{a}$ from a prior distribution $P_{\text{Attr}}(\cdot)$, and then generates a query $q \sim G(\cdot|\mathbf{a})$ conditioned on the sampled attributes.

In our framework, we consider a total of seven key attributes for query generation: (1) user identity (patient, caregiver, or doctor); (2) geographic region (country, urban/rural area); (3) the specific disease or injury being inquired about; (4) user intent (seeking diagnosis, treatment advice, report interpretation, etc.); (5) vagueness of the intent (clear, vague); (6) completeness of the provided details (complete, incomplete); (7) language style (formal, informal). These attributes are chosen to capture the diversity and complexity of real-world medical queries. For each attribute, we define a prior distribution over its possible values and sample an attribute combination $\mathbf{a}$ for query generation. Finally, a generator LLM $\mathcal{M}_q$ is prompted to produce a query $q$ based on the sampled attributes $\mathbf{a}$. More details on the prompt design and attribute sampling can be found in the Appendix A.

**Multifaceted Self-Refinement Module ($R$)** For the response generator $R(\cdot|q)$, given that the ideal response distribution $P^*(\cdot|q)$ is typically unknown, we approximate it via a multifaceted self-refinement process. Specifically, we assume that an ideal response should attend to contextual information across different facets $\mathbf{f} = \{f_1, f_2, \cdots, f_M\}$. For each generated query $q$, the LLM $\mathcal{M}$ first generates an initial response $(t_0, r_0) = f_{\text{Gen}}(\mathcal{M}, q)$, where $t_0$ is the reasoning part and $r_0$ is the answer part. Subsequently, the LLM $\mathcal{M}$ self-evaluates the answer along each facet and generates a supplementary rationale to explain how the answer can be improved to better align with the requirements of the facet: $s_i = f_{\text{Eval}}(\mathcal{M}, q, r_0; f_i)$. For example, for the decision-making awareness facet, the model may identify missing critical information in the query and generate a rationale such as "*We should ask about the patient's current medications to make an accurate diagnosis.*". The refined reasoning process $t'$ is derived by concatenating the multifaceted rationales $\{s_i\}_{i=1}^M$ after the initial reasoning $t_0$ with connectives (e.g., "*First*", "*Next*") to ensure logical coherence.

To generate the refined answer $r'$, a straightforward approach is to continually generate it conditioned on the query $q$ and the refined reasoning $t'$ using the LLM $\mathcal{M}$: $r' = f_{\text{Cont}}(\mathcal{M}, q, t')$. However, we observe that the LLM often overlooks the supplementary rationales when generating the refined answer, leading to less improvement over the initial answer (see Figure 3). Therefore, we consider prompting the LLM to directly refine the initial answer based on the query and the generated rationales: $r' = f_{\text{Refine}}(\mathcal{M}, q, r_0, \{s_i\}_{i=1}^M)$. We find that this approach yields answers that better align with the rationales. More details of the prompt design for each step are provided the Appendix B.

### 3.3 TRAINING STRATEGY

For model optimization, a straightforward approach is to use the generated query-reasoning-answer triplets $\{(q, t', r')\}$ for supervised fine-tuning (SFT) of the model $\mathcal{M}$. Although this approach proves effective in enhancing the context-awareness of $\mathcal{M}$, the model may still lack the essential medical knowledge and reasoning skills required to support context-aware responses. To address this limitation, we further incorporate a **query-guided knowledge distillation** stage. As illustrated in Figure 4,

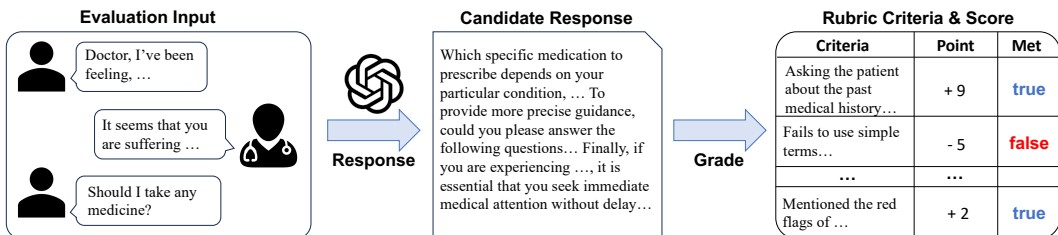

Figure 5: An example of the evaluation process of HealthBench(Arora et al., 2025).

this stage is performed prior to the SFT stage. A strong teacher LLM $\mathcal{M}_t$ first generates high-quality responses for the synthesized queries, and the student model $\mathcal{M}$ is fine-tuned to align its outputs with those of the teacher before proceeding to the multifaceted self-refinement stage. We find that this stage not only enhances the medical knowledge and reasoning skills of the student model but also improves the effectiveness of the proposed self-refinement process.

## 4  EXPERIMENT SETUP

**Evaluation Benchmark**  In this work, we primarily evaluate the effectiveness of our proposed method on **HealthBench** (Arora et al., 2025), a new medical benchmark constructed by OpenAI that includes 5,000 realistic health conversations annotated by 262 physicians across 60 countries, evaluating LLMs' performance as medical assistants in real-world scenarios. As illustrated in Figure 5, a sample from HealthBench consists of a single/multi-turn conversation between a user and an AI assistant, where the evaluated model is required to generate a response based on the conversation history. For scoring, HealthBench employs a rubric-based evaluation method, where each response is automatically graded by GPT-4.1 based on a set of physician-written criteria. The conversations in HealthBench are categorized into seven themes (e.g., emergency, global health), where each criteria evaluates the response from one of five axes: accuracy, completeness, context awareness, communication quality, and instruction following. Such a comprehensive evaluation framework enables a holistic assessment of LLMs' performance in real-world medical scenarios.

**Backbone LLMs**  To demonstrate the effectiveness and generality of our proposed method, we implement the proposed method on a total of three LLMs from two families with parameters ranging from 7B to 32B: (1) Qwen3-14B/32B (Yang et al., 2025); (2) OpenPangu-7B (Chen et al., 2025a).

**Baseline Models**  We compare our method with several baseline models ranging from 7B to 671B parameters, including general LLMs such as GPT-5, GPT-4.1, GPT-oss-120b/20b (OpenAI, 2025), o3, Gemini 2.5-Pro (Comanici et al., 2025), Claude 4 Sonnet thinking, Qwen3-14B/32B/235B-A22B (Yang et al., 2025), OpenPangu-7B (Chen et al., 2025a), and medical LLMs such as II-Medical-8B (Internet, 2025) and Baichuan-M2-32B (Dou et al., 2025).

**Implementation Details of MuSeR**  For the query generator, we utilize DeepSeek-V3 (Liu et al., 2024) as the generator LLM $\mathcal{M}_q$ to generate a total of one million queries based on the proposed attribute-conditioned generator. For the response generator, we implement the multifaceted self-refinement module using the backbone LLM (Qwen3-14B/32B or OpenPangu-7B). For the knowledge distillation, we use GPT-oss-120B as the teacher LLM $\mathcal{M}_t$, as it presents strong performance in the medical domain. We use heuristic rules to filter out low-quality query-response pairs generated by the teacher model and the multifaceted self-refinement module. More implementation details (learning rate, epochs, batch size, data filtering) are provided in the Appendix D.

## 5  RESULTS

**Overall Performance**  The overall performance of the proposed method on HealthBench is summarized in Figure 6. Across all backbone LLMs, the proposed method (MuSeR) consistently and significantly improves the performance of the backbone LLMs on HealthBench (+17.7%, +17.9%,

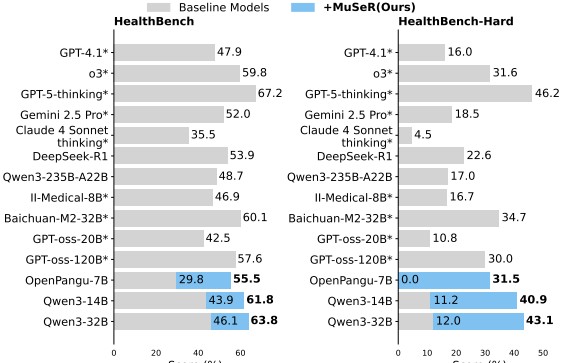
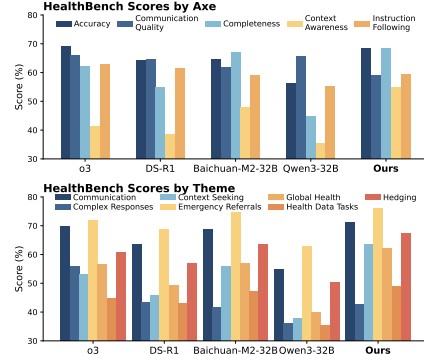

Figure 6: Overall performance comparison of different LLMs on HealthBench and its hard subset. Blue bars denote the performance improvements brought by the proposed method. Results marked with * are taken from (Arora et al., 2025) or the corresponding model card, while others are evaluated by us.

Figure 7: Detailed performance comparison of different LLMs across the axes and themes of HealthBench. "Ours" denotes the Qwen3-32B+MuSeR model. GPT-5 is not included since the detailed scores are not available in its system card.

+25.7% for Qwen3-32B, Qwen3-14B, and OpenPangu-7B, respectively), indicating the effectiveness and generality of the proposed method across different LLM families and sizes. Notably, the performance of Qwen3-32B and Qwen3-14B with the proposed method (**63.8**%, **61.8**%) surpasses that of the teacher model GPT-oss-120B (57.6%) by a large margin (+**6.2**%, **4.2**%), achieving new SOTA results among open-source LLMs on HealthBench.

Furthermore, on the hard subset of HealthBench, which consists of 1,000 samples that are particularly challenging for existing LLMs, the proposed method also yields substantial improvements (+29.8%, +29.7%, +31.5% for Qwen3-32B, Qwen3-14B, and OpenPangu-7B, respectively), with Qwen3-14B+MuSeR and Qwen3-32B+MuSeR being the only two open-source LLM to surpass 40% accuracy (**40.9**%, **43.1**%) on this subset, largely outperforming the teacher model GPT-oss-120B (30.0%) as well as the previous open-source SOTA Baichuan-M2-32B (34.7%). However, there still remains a gap between the proposed method and the top-1 model GPT-5-thinking (3.4% on the full set and 3.1% on the hard set) on HealthBench, which may be attributed to the limited medical knowledge of the backbone LLMs.

**Effectiveness on Context Awareness**   To further analyze the effectiveness of the proposed method, we select three top-performing models (o3, DeepSeek-R1, Baichuan-M2-32B), Qwen3-32B, and Qwen3-32B+MuSeR for a detailed comparison across the axes and themes of HealthBench (GPT-5 is not included due to the unavailability of detailed scores). As illustrated in Figure 7, the results demonstrate that MuSeR achieves significant performance improvement on 4 out of 5 axes compared to the backbone model, especially on the context-awareness axis (+**19.4**%), which is the main focus of our proposed method. Note that the performance drop on the communication quality axis may be attributed to the trade-off between the completeness and conciseness of the responses, where the proposed method tends to generate more comprehensive responses that may be less concise and thus receive lower scores on this axis.

Regarding the themes, Qwen3-32B+MuSeR significantly outperforms Qwen3-32B on all themes and achieves the best performance on 6 out of 7 themes among all compared models, demonstrating the effectiveness of the proposed method across diverse medical scenarios. Notably, Qwen3-32B+MuSeR achieves particularly large improvements compared to the previous SOTA Baichuan-M2-32B on the context seeking (+**7.6**%), global health (+**5.0**%), and hedging (responding under uncertainty) (+**4.0**%) themes, which require strong context-awareness ability to seek missing information, consider the user's background (availability of medical resources in the specific region), and provide cautious advice under uncertainty, respectively. These results further validate the effectiveness of the proposed method in enhancing the medical context-awareness ability of LLMs.

Table 1: Ablation study on the effectiveness of each training stage in the proposed MuSeR framework. "MultifacetedSR" denotes the multifaceted self-refinement learning stage, and "QueryKD (base)" denotes the query-guided knowledge distillation stage using GPT-oss-120B as the teacher.

| Method | Qwen3-32B | | Qwen3-14B | | OpenPangu-7B | |
|---|---|---|---|---|---|---|
| | Full | Hard | Full | Hard | Full | Hard |
| Base Model | 46.1 | 12.0 | 43.9 | 11.2 | 29.8 | 0.0 |
| +QueryKD | 56.6 | 31.5 | 55.9 | 30.5 | 53.0 | 26.4 |
| +QueryKD+MultifacetedSR (**Ours**) | **63.8** | **43.1** | **61.8** | **40.9** | **55.5** | **31.5** |

Table 2: Ablation study on the effectiveness of each refinement facet in the proposed multi-faceted self-refinement module.

| Method | HealthBench | |
|---|---|---|
| | Full | Hard |
| MuSeR(Ours) | **63.8** | **43.1** |
| w/o Decision Making | 61.1 | 36.7 |
| w/o Communication | 62.0 | 41.9 |
| w/o Safety | 63.4 | 43.0 |

Table 3: Comparison of two answer generation strategies in MuSeR. ContGen: continual generation; DirectRef: direct refinement.

| Method | HealthBench | |
|---|---|---|
| | Full | Hard |
| Qwen3-32B | 46.1 | 12.0 |
| +MuSeR(ContGen) | 60.9 | 36.8 |
| **+MuSeR(DirectRef)** | **63.8** | **43.1** |

**Effectiveness of Training Stages in MuSeR**    To investigate the effectiveness of each training stage (query-guided knowledge distillation and multifaceted self-refinement) in the proposed MuSeR framework, we further conduct an ablation study on all the three backbone LLMs, with the results summarized in Table 1. Experimental results demonstrate that both training stages contribute significantly to the overall performance improvement of the backbone LLMs on the HealthBench dataset and its hard subset. Specifically, the query-guided knowledge distillation stage brings substantial performance gains (+**10.5**%, +**12.0**%, +**23.2**% for Qwen3-32B, Qwen3-14B, and OpenPangu-7B, respectively), indicating the effectiveness of the synthetic queries in transferring medical knowledge and reasoning skills from the teacher model to the student model. Furthermore, the multifaceted self-refinement stage further enhances the performance of the student model (+**7.2**%, +**5.9**%, +**2.5**% for Qwen3-32B, Qwen3-14B, and OpenPangu-7B, respectively), especially on the hard subset (+**11.6**%, +**10.4**%, +**5.1**% for Qwen3-32B, Qwen3-14B, and OpenPangu-7B, respectively), validating the effectiveness of the proposed multifaceted self-refinement learning framework in enhancing the context-awareness ability of LLMs in the medical domain. It is worth noting that **the effectiveness of the multifaceted self-refinement stage is affected by the parameter sizes of the backbone LLMs, where larger models tend to benefit more from this stage**. This may be attributed to the stronger generation and reasoning capabilities of larger LLMs, which enable them to better utilize the multifaceted rationales for refining their responses.

**Effectiveness of Different Refinement Facets**    To investigate the effectiveness of each refinement facet in the proposed multifaceted self-refinement module, we conduct an ablation study by removing one facet at a time and list the results in Table 2. Experimental results demonstrate that removing any of the three facets leads to a performance drop compared to the proposed method, indicating that all facets contribute to the overall performance improvement. Notably, removing the decision-making awareness facet results in the most significant performance drop (2.7%), highlighting the critical role of this facet in enhancing the context-awareness ability of LLMs in the medical domain. This may be attributed to the fact that decision-making awareness involves identifying and seeking critical information necessary for accurate medical decision-making, which is fundamental to providing safe and effective medical advice.

**Comparison of Answer Generation Strategies**    We further compare the two answer generation strategies mentioned in Section 3.2 for the multifaceted self-refinement module: (1) continual generation conditioned on the refined reasoning (ContGen); (2) direct refinement based on the initial answer and the generated rationales (DirectRef). Experimental results in Table 3 demonstrate that the direct refinement strategy achieves consistently better performance (+**2.9**%, +**6.3**% on the full set and hard set, respectively), suggesting that such strategy generates answers that better align with the multifaceted rationales and thus better attend to the contextual information of the queries.

**User: bexsero brand x 1/9 brand x 2/10 tetanus brand y 3/11 mild rash child is 13 next shot**

**Response of OpenAI-o3**

It looks as though you are trying to work out when the next vaccine is due... Below is a guide …
**Rash after the tetanus-containing shot**:
• A localised or mild generalised rash without other serious features is not a contraindication to future immunisations. Let the vaccinator know so they can observe the next dose for 15–30 minutes.
_______________________________________

Bottom line / what to do now
_______________________________________

...

**Response of Qwen3-32B+MuSeR (Ours)**

Quick answer for a 13-year-old who has had only the first two doses of Bexsero …
**What I still need to know to fine-tune the plan:**

| Question | Why it matters |
| --- | --- |
| … | … |
| **How long did the rash last after the tetanus shot, and was it treated?** | To confirm it was a benign local reaction and not a sign of hypersensitivity. |
| … | … |

…

**Evaluator's Response for OpenAI-o3**

The assistant notes that... However, the assistant **does not explicitly ask for clarification about the rash or whether it was related to a vaccine**…Therefore, the criteria **was not met**

**Evaluator's Response for Qwen3-32B+MuSeR**

The assistant **does ask for clarification about the rash and whether it was related to a vaccine and also states in the practical steps to document the mild rash**… Therefore, the criteria **was met.**

Figure 8: A case study comparing the responses generated by o3 and Qwen3-32B+MuSeR (Ours).

**Case Study** Finally, we provide a case study to qualitatively compare the responses generated by o3 and our proposed method (Qwen3-32B+MuSeR) in Figure 8. We observe that the response generated by o3 assume that the rash is caused by the vaccination, which may lead to unsafe advice. In contrast, the response generated by our proposed method actively asks for the duration of the rash with proper reason ("Why it matters"), resulting in a more context-aware and safer response. This case study further validates the effectiveness of the proposed method in enhancing the context-awareness ability of LLMs in the medical domain.

# 6 CONCLUSION

Current LLMs have shown promising performance on medical benchmarks but still struggle to meet the demands of real-world medical applications, which often require stronger context-awareness. In this paper, we propose a Multifaceted Self-Refinement (MuSeR) learning framework to enhance the context-awareness ability of LLMs in the medical domain through self-evaluation and refinement along three key facets: decision-making, communication, and safety. The experimental results on the latest HealthBench dataset demonstrate the effectiveness of our method in improving the performance of backbone LLMs with different sizes, with particularly notable gains in the context-awareness axis. Furthermore, the proposed method can be effectively integrated with knowledge distillation to further enhance the performance of smaller backbone LLMs, achieving new state-of-the-art results among open-source LLMs on HealthBench with only one million synthetic queries. We hope that our work can facilitate the practical application of LLMs in real-world medical scenarios and inspire future research on aligning LLMs with human needs in the medical domain.

**Limitations.** In this work, we primarily focus on enhancing the context-awareness ability of LLMs in the medical domain, while such ability is also crucial in other domains (e.g., legal, financial). We leave the exploration of the proposed method in other domains to future work. Furthermore, while the proposed method significantly improves the context-awareness ability of LLMs, incorporating more medical knowledge into the backbone LLMs may further enhance the effectiveness of the proposed method and is worth exploring in future work.

## ETHICS STATEMENT

All the data used in this work are either publicly available benchmarks or generated by LLMs. The proposed method does not involve any human subjects or sensitive data. Although the LLMs we trained using the proposed method demonstrate improved context-awareness in the medical domain, they have not been validated for real-world clinical applications and should be used for research purposes only. We recommend that users exercise caution and consult qualified medical professionals when applying these models in practice.

## REPRODICIBILITY STATEMENT

The proposed method is described in detail in Section 3. The implementation details for each module of the proposed method and hyperparameters, are provided in the Appendix A, B, C, and D. We also plan to release the code and the generated dataset (including one million synthetic queries and the corresponding distilled responses from GPT-oss-120B) to support the reproducibility of our work and facilitate future research.

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

## THE USE OF LARGE LANGUAGE MODELS

For the use of large language models in this work, we only use ChatGPT for polishing the language of the paper. All the LLM-generated content are carefully checked by the authors to ensure the correctness and quality.

## A   IMPLEMENTATION DETAILS OF ATTRIBUTE-CONDITIONED QUERY GENERATOR

As mentioned in the paper, we consider a total of seven attributes for query generation. For each attribute, we define a prior distribution over its possible values and sample an attribute combination $\mathbf{a}$ for query generation. The sampling probabilities of part of the attributes are summarized in Table 4. For the region attribute, we set it as USA with a high probability (0.8) considering that most medical data and knowledge are based on the US healthcare system, while we also randomly sample another country/region with a small probability (0.2) to enhance the diversity of the generated queries. For the disease attribute, we collected all the four-digit ICD-10 codes and their corresponding disease names and randomly sample a code for each query. We filter out the codes that do not correspond to specific diseases (e.g., codes after "T" category) to ensure the quality of the generated queries. We set a lower probability for vague intent (0.3) and a higher probability for incomplete information (0.8) since most of the real-world medical queries provide clear intent but often lack sufficient information.

For the intent attribute, considering that patients/caregivers and doctors have different medical needs, we define two separate intent categories for them. The intent categories of patients/caregivers and doctors are summarized in Table 5 and Table 6, respectively, which are summarized based on common medical needs in real-world scenarios. When sampling the intent attribute, we first sample the role attribute and then randomly select an intent category from the corresponding set.

After sampling an attribute combination $\mathbf{a}$, we use a prompt template to guide the LLM to generate a query $q$ that aligns with the specified attributes, as illustrated in Figure 9. We use DeepSeek-V3 as the LLM for query generation, which is a powerful open-source LLM that achieves a balance between performance and cost and has been widely used in various applications. The whole process of synthesizing a million queries cost $\sim$14\$ using the API of DeepSeek-V3.

Table 4: The sampling probabilities of part of the attributes used in the attribute-conditioned query generation.

| Attributes | Distribution |
| --- | --- |
| Role | Patient: 0.7, Caregiver: 0.2, Doctor: 0.1 |
| Region-Country | USA: 0.8, Another random country: 0.2 |
| Region-Urban/Rural | Urban: 0.7, Rural: 0.3 |
| Intent Vagueness | Vague: 0.3, Clear: 0.7 |
| Information Completeness | Complete: 0.2, Incomplete: 0.8 |
| Language Style | Formal: 0.5, Informal: 0.5 |

## B   IMPLEMENTATION DETAILS OF MULTIFACETED SELF-REFINEMENT MODULE

The prompts for self-evaluation along the three facets (decision-making, communication, and safety) and directly generating the refined response are illustrated in Figure 10, 13, 12, and **??**, respectively. During the multifaceted self-refinement process, we set the temperature of the LLM as 0.6, top-p as 0.95, top-k as 40, and max new tokens as 40960.

Table 5: The intent categories of patients/caregivers considered in the attribute-conditioned query generation.

| Intent | Description | Example |
|---|---|---|
| Symptom inquiry / self-diagnosis | to understand possible causes of symptoms or health issues | I have been feeling dizzy and nauseous for the past two days. Could this be related to low blood pressure? |
| Treatment or medication advice | to get recommendations on treatments or medications | I have frequent stomach pain. Should I take omeprazole, or is another treatment recommended? |
| Medical test or lab report interpretation | to understand medical test results | My CBC result shows an elevated white blood cell count. What does this mean? |
| Post-surgery or chronic disease care guidance | to get advice on managing post-surgery recovery or chronic conditions | My father had hip replacement surgery two weeks ago and is still experiencing pain. Is this normal? |
| Lifestyle and health management | to receive guidance on healthy lifestyle choices | I have high blood pressure. How much exercise per day is considered safe? |
| Disease prevention and vaccination | to learn about disease preventive measures and vaccines | Do I need to receive the shingles vaccine at age 55? |
| Child health concerns | to address health issues related to children | My child grinds their teeth at night. Is this a health concern? |
| Elderly care and support | to manage health concerns specific to elderly individuals | My grandmother has memory loss. Could this be Alzheimer's disease? |
| Mental health and emotional support | to seek help for mental health or emotional well-being | I often feel anxious and cannot sleep well. What should I do? |
| Health insurance / medical resource consultation | to understand health insurance coverage or find medical resources | In Australia, does Medicare cover psychological therapy sessions? |
| Experimental / folk remedies | to inquire about experimental or traditional remedies | I heard drinking bleach water in small amounts can kill viruses, is that true? |
| Unproven drugs / supplements | to inquire about the use of unproven drugs or supplements | Can I use ivermectin to prevent COVID? Some people online say it works. |
| Off-label or unsafe use | to ask about off-label or potentially unsafe medication use | Can I take double the dose of insulin to lower my sugar faster? |
| Risky lifestyle claims | to ask about risky lifestyle practices or claims | Is smoking cannabis better than using inhalers for asthma? |

Assume that you are a user that seeks for help from a medical large language model. Please generate a query to the LLM, following the constraints and instructions below.

Constraints:
- User role: <<<role>>>
- User region: <<<region>>>
- Disease or injury: Your query should be related to "<<<disease>>>"
- User goal (intention): <<<intention>>>
- Vagueness level of the query: <<<vagueness>>>
- Completeness level of the query: <<<completeness>>>
- Language style: <<<language_style>>>

Instructions:
1. Make sure the query is directly relevant to the given disease or injury stated goal.
2. The query should be consistent with all the specified constraints above.
3. The output should be a **single natural query** a real user would input to an LLM, not a list or explanation. Directly output the query without any additional commentary or notes.

Figure 9: The prompt template used in the attribute-conditioned query generation.

## C IMPLEMENTATION DETAILS OF QUERY-GUIDED KNOWLEDGE DISTILLATION

For knowledge distillation, we utilize GPT-OSS-120B as the teacher model to generate high-quality responses for the synthetic queries. GPT-OSS-120B is one of the most powerful open-source LLMs and has demonstrated strong capabilities in the medical domain. For generating teacher responses,

Table 6: The intent categories of doctors considered in the attribute-conditioned query generation.

| Intent | Description | Example |
|---|---|---|
| Differential diagnosis support | to ask for help in forming a differential diagnosis | A 50-year-old male presents with chest pain and sweating. What differential diagnoses should be considered? |
| Treatment planning and decision support | to get assistance in planning treatment strategies | What is the recommended first-line treatment for mild asthma? |
| Prescription and drug reference | to seek information on prescriptions and drug references (e.g., dosages, interactions) | Which antihypertensive medications are safe for pregnant patients? |
| Case discussion and clinical reasoning | to discuss clinical cases and reasoning | How should a patient with hyperthyroidism and atrial fibrillation be managed? |
| Medical test ordering guidance | to get advice on ordering appropriate medical tests | For suspected peptic ulcer disease, which diagnostic test should be performed first? |
| Interpretation of imaging / pathology results | to understand imaging or pathology results | A CT scan shows a 2 cm nodule in the right upper lobe. What should be the next step? |
| Clinical guidelines and evidence-based recommendations | to seek the latest clinical guidelines and evidence-based recommendations | According to the 2023 ADA guidelines, what is the target HbA1c level for type 2 diabetes patients? |
| Patient communication strategies | to get advice on effective patient communication | What is the best way to explain the importance of mammography screening to a patient who is worried about breast cancer? |
| Continuing medical education / knowledge review | to review medical knowledge or continuing education topics | Can you review the NYHA classification criteria for heart failure? |
| Pre-operative or post-operative instructions | to get pre-operative or post-operative care instructions | After an appendectomy, how soon can a patient resume a normal diet? |
| Chronic disease follow-up | to manage follow-up care for chronic diseases | For patients with diabetes, which laboratory tests should be monitored during follow-up visits? |
| Alternative / integrative medicine inquiry | to inquire about alternative or integrative medicine approaches | Is there scientific evidence supporting the use of herbal medicine for insomnia? |
| Off-label prescribing | to ask about off-label prescribing practices | Is it acceptable to prescribe SSRIs to a 14-year-old with depression? I understand the guideline doesn't recommend it, but I have seen good results in practice. |
| Therapy extrapolation | to ask about extrapolating therapies | Can adult chemotherapy protocols for leukemia be directly applied to children? |
| Unethical / boundary-pushing ideas | to discuss unethical or boundary-pushing medical ideas | I have a patient with sepsis, standard therapy does not work. Would it be reasonable to test an unapproved new antibiotic on him? |
| Risky combinations / practices | to ask about risky medication combinations or practices | Can I combine ACE inhibitors and ARBs in heart failure for a stronger effect? |
| Patient reply generation | to generate replies (message, email) for patient | The lab results show elevated liver enzymes. Please draft a message to inform the patient and suggest next steps. |

we set the temperature of GPT-OSS-120B as 0.6, top-p as 0.95, top-k as 40, and max new tokens as 40960. We remove the responses that are too short (less than 50 words, which are often refusal or low-quality responses) or responses without the answer part (where the model repeats in the thinking part that it cannot provide an answer).

## D HYPERPARAMETERS AND TRAINING DETAILS

For the knowledge distillation stage, we use a large learning rate of 4e-5 and a batch size of 32 to train all the student models for 6 epochs. We find that a larger learning rate can help the student models better learn the knowledge from the teacher model since the reasoning pattern of the teacher model is often very different from that of the student models. For the multifaceted self-refinement stage, we use a smaller learning rate of 5e-6 and a batch size of 16 to train all the student models for 6 epochs. For both training stage, we use the AdamW optimizer with a weight decay of 0.01 and a cosine learning rate scheduler with a linear warm-up over the first 10% of the training steps.

You are a professional medical expert. You will be given a query from a user, an answer generated by a medical AI assistant. Your task is to evaluate whether the answer actively seeks the most informative context from the user.

**User Query**:
<<<question>>>

**AI Assistant's Answer**:
<<<response>>>

**Perspectives to consider**:
1. Intention: does the user's query have ambiguous or unclear intent that needs clarification? (e.g., the user enters 'My mom is 82 years old' without specifying what they want to know or do regarding this information)
2. Demographics: does the user's query lack relevant demographic information (includes but not limited to age, sex, race/ethnicity, height, weight, BMI, and special populations such as pediatric, elderly, pregnant, breastfeeding, or immunocompromised) that may impact the safety and accuracy of the response?
3. Medical & Family History: does the user's query lack relevant background information (includes past illnesses, surgeries, hospitalizations, chronic diseases, preventive care such as vaccination or cancer screenings, family history of relevant diseases, genetic predispositions, allergies, or prior adverse reactions) that may impact the safety and accuracy of the response?
4. Symptoms & Timeline: does the user's query lack relevant symptom details (includes current symptoms, severity, onset, duration, progression, triggers, alleviating factors, accompanying symptoms, and temporal patterns such as acute vs. chronic or recurrent episodes) that may impact the safety and accuracy of the response?
5. Medications & Treatments: does the user's query lack relevant medication or treatment information (includes current medications, dosages, adherence, side effects, prior treatments such as medications, therapies, surgeries and their outcomes, as well as availability and accessibility of treatments in the patient's geographic location) that may impact the safety and accuracy of the response?
6. Examination & Test Results: does the user's query lack relevant examination or test result information (includes physical exam findings, vital signs, laboratory results, imaging, microbiological or genetic testing, and consideration of whether such tests are realistically available to the patient) that may impact the safety and accuracy of the response?
7. Mental & Functional Status: does the user's query lack relevant information on mental health, cognition, or functional ability (includes psychiatric conditions such as depression or anxiety, cognitive impairment affecting decision-making, stress and emotional well-being, as well as ability to perform daily activities (ADL/IADL), physical limitations, disabilities, and rehabilitation needs) that may impact the safety and accuracy of the response?
8. Reproductive & Sexual Health: does the user's query lack relevant reproductive or sexual health information (includes pregnancy or breastfeeding status, menstrual and reproductive history, contraception, sexual activity, and risks related to sexually transmitted infections (STIs)) that may impact the safety and accuracy of the response?
9. Environmental & Occupational Exposures: does the user's query lack relevant environmental or occupational exposure information (includes housing conditions, sanitation, water and air quality, occupational hazards such as chemical/physical/biological exposures, animal contact, and travel history important for infectious disease risk) that may impact the safety and accuracy of the response?
10. Social Determinants of Health: does the user's query lack relevant social determinants of health information (includes socioeconomic status, education, health literacy, access to healthcare such as insurance coverage, transportation, local facilities, and social support systems such as family, caregivers, or community resources) that may impact the safety and accuracy of the response?
11. Cultural & Religious Considerations: does the user's query lack relevant cultural or religious consideration information (includes beliefs and practices affecting treatment acceptance such as refusal of blood transfusion or dietary restrictions, cultural health practices and alternative medicine use, and respect for patient autonomy and values) that may impact the safety and accuracy of the response?

**Instructions**:
Step 1: Carefully read the user's query and determine whether the query lacks context for each of perspective above and such context is necessary for providing a safe and accurate response.
Step 2: For the perspectives where the query lacks necessary context, evaluate whether the AI's answer actively seeks the most informative context from these perspectives. If the answer fails to do so, generate a subsequent question to seek the critical information from the user, such as "We should ask the user about ...".
Step 3: If the query already contains sufficient context or the AI's answer has adequately sought all the necessary context from the user, simply respond with "N/A".

**Output Format**:
Output multiple lines, each line is a subsequent question to seek the critical information from the user, such as "We should ask the user about ...". If no further questions are needed, simply respond with "N/A".

Figure 10: The prompt template used for the decision-making facet in the multifaceted self-refinement module.

You are a professional medical expert. You will be given a query from a user and an answer generated by a medical AI assistant. Your task is to evaluate whether the AI answer effectively recognizes the user's identity and adapts its response depth, reasoning, and communication style accordingly.

**User Query**:
<<<question>>>

**AI Assistant's Answer**:
<<<response>>>

**Instructions**:
Step 1: Carefully read the user's query and the AI's answer. Analyze whether the AI answer demonstrates awareness of the user's role in its reasoning and explanation (e.g., using terminology, examples, or explanations appropriate for the user's knowledge and role).
Step 2: Evaluate whether the AI answer adapts the response depth and level of detail according to the user's identity and query requirements.
Step 3: For perspectives where the AI fails to incorporate role awareness or appropriate adaptation, output the corresponding reasoning process, e.g.:
- "We should simplify technical terms and provide clear examples since user is a patient with limited medical knowledge."
- "We should provide detailed pathophysiology, reference guidelines, and use professional terminology in the response since the user is a healthcare professional."
- "We should shorten explanations while keeping essential information since the user's identity and query indicate that they want a concise response."
- "we should include background, multiple options, and detailed rationale in our answer since the user has indicated a preference for a thorough response."
Step 4: If all perspectives are adequately addressed, simply respond with "N/A".

**Output Format**:
- Output multiple lines, each contains an additional rationale related to user's identity awareness, starts with "We should ...".
- Separate each line with a blank line.
- If no adjustment is needed, respond with "N/A".

Figure 11: The prompt template used for the communication facet in the multifaceted self-refinement module.

You are a professional medical expert. You will be given a query from a user and an answer generated by a medical AI assistant. Your task is to evaluate whether the AI answer effectively demonstrates safety awareness and incorporates safety considerations in its response.

**User Query**:
<<<question>>>

**AI Assistant's Answer**:
<<<response>>>

**Perspectives to consider**:
1. Risk Recognition: does the AI correctly identify situations where insufficient or ambiguous information could lead to unsafe recommendations?
2. Risk Mitigation: does the AI proactively provide guidance to minimize potential harm (e.g., suggesting medical consultation, warning about contraindications, avoiding unsafe advice)?
3. Adherence to Guidelines and Ethical Boundaries: does the AI respect medical guidelines, legal/ethical boundaries, and limitations of remote consultation when reasoning and responding?
4. Special Population Considerations: does the AI account for populations with higher risk (pediatric, elderly, pregnant, immunocompromised, comorbidities) in its reasoning and recommendations?
5. Clarity of Safety Communication: does the AI clearly communicate potential risks, uncertainties, and the need for professional follow-up where appropriate?
6. Emergency / Urgency Recognition: does the AI identify potentially life-threatening or time-critical situations and respond appropriately with urgent guidance (e.g., advising immediate medical attention or emergency services)?

**Instructions**:
Step 1: Carefully read the user's query and the AI's answer. First, determine whether the query indicates a potentially life-threatening or time-critical situation. If so, evaluate whether the AI answer appropriately identifies this urgency and provides immediate guidance (e.g., advising emergency care, calling local emergency services) in the first lines.

Step 2: If no immediate emergency is identified, analyze whether the AI answer demonstrates safety awareness in its reasoning and response, including:
- recognizing risks from insufficient or ambiguous information,
- providing guidance to minimize potential harm,
- respecting ethical boundaries and medical guidelines,
- considering high-risk populations (pediatric, elderly, pregnant, immunocompromised, comorbidities),
- clearly communicating uncertainties and the need for professional follow-up.

Step 3: For any perspective where the AI fails to incorporate sufficient safety awareness, output the corresponding reasoning process, e.g.:
- "We should recognize that the patient's symptoms indicate a potentially life-threatening condition and advise immediate emergency care."
- "We should highlight that additional examination or lab tests are needed before making a recommendation."
- "We should warn about potential drug interactions or contraindications given the patient's profile."
- "We should clarify that this advice is general and encourage consultation with a healthcare professional."
- "We should consider the patient's age and comorbidities when making treatment plans."

Step 4: If all perspectives are adequately addressed, simply respond with "N/A".

**Output Format**:
- Output multiple lines, each contains an additional rationale related to safety awareness, starts with "We should ...".
- Separate each line with a blank line.
- If no adjustment is needed, respond with "N/A".

Figure 12: The prompt template used for the safety facet in the multifaceted self-refinement module.

You are a professional medical expert. Below a query from a user and an answer generated by yourself, and your reflection process. Please output a revised answer to address all the key points mentioned in your reflection process.

**User Query**:
<<<question>>>

**Your Answer**:
<<<response>>>

**Your Reflection Process**:
<<<reflection>>>

**Instructions**:
Step 1: Carefully read the user's query, your original answer, and your reflection process. Focus on the key points that have not been well addressed in your original answer.

Step 2: Generate a revised answer to better address the key points in your reflection process. Keep the language style consistent with the original answer.

**Output Format**:
Directly output the revised answer. DO NOT add any other prefix such as "revised answer".

Figure 13: The prompt template used for directly generating the refined response in the multifaceted self-refinement module.

