# OpenReview forum: "Enhancing the Medical Context-Awareness Ability of LLMs via Multifaceted Self-Refinement Learning"
_ICLR.cc/2026/Conference — Submitted to ICLR 2026_

### Official Review · Reviewer_gMLw · 2025-11-01

**Soundness:** 2
**Presentation:** 2
**Contribution:** 2
**Rating:** 2
**Confidence:** 4

**Summary:**

This paper proposes MuSeR (Multifaceted Self-Refinement), a data-driven approach that enhances an LLM's context-awareness by (1) generating diverse, attribute-conditioned queries, (2) having the LLM self-evaluate and refine its own answers along three facets (decision-making, communication, and safety), and (3) using these refined pairs for supervised fine-tuning.

**Strengths:**

1. The paper convincingly argues that limited context-awareness impedes real-world medical deployment and positions MuSeR as a principled remedy.
2. The attribute-conditioned query generator is a practical way to overcome real-world data scarcity while stress-testing models across user roles, regions, intents, and ambiguity levels.

**Weaknesses:**

1. Notation is difficult to follow. Consolidate all symbols in a notation table; unify subscripts/superscripts
2. Facet selection lacks empirical justification. Provide evidence for focusing on Decision-Making, Communication, and Safety (e.g., error taxonomy from baseline models, clinician interviews), and include an ablation showing the marginal gain of each facet and their interactions.
3. In Line 236, why "we consider a total of seven key attributes for query generation"?
4. Dependence on query-guided knowledge distillation (KD). Include an ablation of Base Model + MuSeR (no KD) versus MuSeR + KD to quantify KD’s incremental benefit; report compute/cost trade-offs and performance deltas.

**Questions:**

1. Why were the three facets and seven attributes chosen?
2. What ablations isolate MuSeR’s gains from knowledge distillation and refinement iterations, and how does it generalize across domains/OOD with cost/latency reported?

---

> ### Author Response · Authors · 2025-12-03
> **Response to Reviewer gMLw**
>
> 1. **Issue of Notation**: Thank you for your feedback. In fact, the notation in our paper follows standard conventions in the machine learning field, and we have provided clear definitions whenever a new symbol is introduced. In addition, we have illustrated key implementation details of our method with corresponding figures (Figures 2, 3, and 4) to further help readers understand the content. We would like to understand which part of the paper you found difficult to follow in terms of notation, and could you also clarify which part you are referring to regarding "unifying subscripts/superscripts"?
>
> 2. **Justification of Facet Selection**: Thank you. The motivation of this work is that although existing LLMs generally achieve considerable performance on medical benchmarks, they still perform poorly in real-world medical scenarios (e.g., medical conversations), particularly due to the deficiencies in identifying key medical contextual information. In this paper, we focus on key facets of medical context-awareness by considering the evaluation axes of the authoritative HealthBench [1], and also consider the shortcomings in medical capabilities exposed by several medical evaluation studies. Based on these insights, we select **decision-making**, **communication**, and **safety** as three key medical context-awareness facets:
>
>    **Decision-making**: Decision-related contextual information (e.g., patient demographics, symptoms, medications, medical history) is essential for a medical assistant to provide helpful responses. Existing medical benchmarks typically provide all necessary decision-making information within the prompt, and LLMs tend to perform well under such conditions. However, in real-world settings (e.g., patient consultations), users rarely provide complete decision-related information at the outset (either due to missing diagnostic tests or simply not knowing what information is relevant for further decision-making). However, current LLMs often struggle to provide helpful responses when initial decision-making information is insufficient [2–3]. Therefore, we include decision-making as one dimension of context-awareness, aiming to enhance the model’s ability to proactively request missing information under incomplete initial contexts.
>
>    **Communication**: Communication-related contextual information (e.g., user identity, geographic location) is also crucial for generating helpful medical responses. This includes perceiving the user’s identity (e.g., patient vs. clinician) to adjust the level of medical terminology and detecting location information to reason about local resource availability (e.g., certain diagnostic tests may not be accessible in specific regions). Effective perception of such contextual cues is fundamental for providing personalized medical advice.
>
>    **Safety**: The medical domain adheres to the strict principle of non-maleficence, meaning that providing safe medical guidance is a core requirement for any medical assistant. To ensure safety, a model must be able to identify potential risk factors in user inputs and alert users to possible hazards. However, prior work [4] shows that existing LLMs often lack an understanding of the relative weight and priority of risk factors, which leads to difficulties in handling complex and uncertain real-world clinical scenarios. Therefore, we consider the effective perception of risk factors a key dimension of medical context-awareness as well.
>
>    In addition, regarding the ablation study you mentioned, we have already presented the relevant experimental results in **Table 2** of the paper. Therefore, we are not quite sure what additional form of ablation study you would like to see to further validate the rationale for our selected facets.

---

> > ### Author Response · Authors · 2025-12-03
> >
> > 3. **"Why 'we consider a total of seven key attributes for query generation'"**: Thank you for your comments. In our query generator, we consider a total of **seven key attributes**:
> >
> >       1. **User identity** (patient, caregiver, or doctor)
> >       2. **Geographic region** (country, urban/rural area)
> >       3. **Specific disease or injury being inquired about**
> >       4. **User intent** (seeking diagnosis, treatment advice, report interpretation, etc.)
> >       5. **Vagueness of the intent** (clear, vague)
> >       6. **Completeness of the provided details** (complete, incomplete)
> >       7. **Language style** (formal, informal)
> >
> >       These attributes are closely related to the three context-awareness facets we consider:
> >
> >       - **Communication**: user identity, geographic region, and language style. The model needs to adjust the level of professionalism in responses according to user identity and language style, while also considering the availability of medical resources in the user’s region.
> >       - **Safety**: disease/injury and user intent. The model needs to pay attention to potential risk factors associated with specific diseases and be able to refuse unsafe user requests (e.g., taking unverified therapies).
> >       - **Decision-making**: vagueness of intent and completeness of provided details. The model needs to actively inquire about necessary decision-making information when the user’s intent is unclear or key information is missing.
> >
> >       Indeed, real-world medical queries depend on more factors than these seven attributes. Nevertheless, our experiments show that queries constructed based on these attributes already exhibit substantial diversity, and performing multifaceted self-refinement learning on these queries can effectively enhance the model’s context-awareness capabilities.
> >
> > 4. **Effectiveness of query-guided knowledge distillation (KD)**： Thank you for your comments. We would like to emphasize that our proposed Multifaceted Self-Refinement method **does not** depend on the knowledge distillation module. The role of knowledge distillation is simply to inject missing medical knowledge into smaller models, thereby further enhancing the improvement achieved by our method. To illustrate this point, we directly applied Multifaceted Self-Refinement learning to Qwen3-32B without any knowledge distillation. The results are as follows:
> >
> > | Method                            | HealthBench |
> > | --------------------------------- | ----------- |
> > | Qwen3-32B                         | 46.1        |
> > | +MultifacetSR                     | 48.9        |
> > | +QueryKD                          | 56.6        |
> > | **+QueryKD+MultifacetSR (MuSeR)** | **63.8**    |
> >
> > The experimental results show that Qwen3-32B+MultifacetSR also achieves a 2.8% improvement compared to the backbone model. After introducing QueryKD, the improvement brought by Multifaceted Self-Refinement further increases from 2.8% to 7.2%. This demonstrates that knowledge distillation can effectively supplement missing medical knowledge in smaller models, thereby further enhancing the effectiveness of Multifaceted Self-Refinement learning.
> >
> > Regarding your comment on "reporting compute/cost trade-offs and performance deltas," we are not entirely sure what specific compute/cost trade-offs and performance deltas you are referring to, nor how they relate to knowledge distillation. Would you mind further clarifying these so I can further address these concerns?
> >
> > 5. **"What ablations isolate MuSeR’s gains from knowledge distillation and refinement iterations, and how does it generalize across domains/OOD with cost/latency reported?"**: We are very sorry, but we feel it difficult to fully grasp your question. In fact, our paper and method do not involve any "multiple iterations", and therefore we cannot report any "gains from refinement iterations." Additionally, we would like to kindly remind that this work is specifically focused on addressing the key issue of insufficient medical context-awareness in existing LLMs, rather than general-domain problems. Of course, we do not rule out the possibility that our proposed method could be applied to other domains in the future to enhance context-awareness; however, this is beyond the scope of the current study.

---

### Official Review · Reviewer_Y1NA · 2025-11-01

**Soundness:** 3
**Presentation:** 3
**Contribution:** 2
**Rating:** 4
**Confidence:** 3

**Summary:**

Authors propose MuSeR, a Multifaceted Self-Refinement framework designed to enhance LLMs’ context-awareness in medical tasks. Proposed framework first generates synthetic data by simulating real-world medical queries with controlled attributes (e.g., user role, region, disease, intent). It then performs self-refinement of the model’s responses along three key facets: (i) decision-making, (ii) communication, and (iii) safety. The refined dat used for supervised fine-tuning (SFT) and can optionally be combined with knowledge distillation from a larger teacher model (GPT-oss-120B). They evaluate their framework on HealthBench, where MuSeR achieves top performance using Qwen3-14/32B, even surpassing its teacher model and other much larger proprietary models.

**Strengths:**

- Authors point-out a well-motivated and timely issue in the medical LLM domain by emphasizing the distinction between exam-style benchmarks and real-world medical scenarios., where they focus on context-awareness makes the task notably harder yet more realistic and relevant for clinical applications.
- According to the authors’ results, MuSeR outperforms several top-priority models and achieves closer performance with gpt-5-thinking while using much smaller backbones (e.g., Qwen3-14B). Interestingly, it even surpasses its teacher model (gpt-oss-120b), which is a counterintuitive yet noteworthy finding that can be valuable for future research.

**Weaknesses:**

- The main limitation is the narrow evaluation setup. Although the authors conduct experiments with different model families and baselines for comparison, they evaluate solely on HealthBench, making it difficult to assess the framework's robustness. It would be beneficial to see results on at least 2-3 additional benchmarks. Moreover, HealthBench relies on GPT-4.1 as an LLM-as-a-Judge; such evaluations require statistical consistency tests, where the same experiments should be run multiple times with different LLMs, reporting alignment across models and standard deviations within the same model to better understand robustness.
- Catastrophic forgetting is a common problem in post-training domain-specific LLMs and is not addressed in the current manuscript. Since the authors conduct heavy fine-tuning on generated data, is there any evidence of catastrophic forgetting; if not how they overcome that?
- Finally, the "self-refinement" terminology seems confusing, as the proposed framework also includes a distillation step. Why do the authors frame their approach as self-refinement, and could they provide more details?

**Questions:**

1. Could the authors share a reference for lines 42-43?
2. The provided repository seems to include only the readme file.
3. In lines 192-193, the authors mention that “the formulations above represent our design goals rather than explicit optimization objectives.” What does this mean, and why formulate equations that are not followed in the framework?
4. It is interesting that a student model can outperform the teacher model; could the authors share theoretical insights on this?

---

> ### Author Response · Authors · 2025-12-03
> **Response to Reviewer Y1NA**
>
> We sincerely appreciate your thorough review. Below, we address each of your concerns in detail.
>
> 1. **Evaluation Setup Issue**: We sincerely appreciate your insightful suggestions. The motivation behind our work arises from the observation that, although existing LLMs achieve strong performance on standard medical benchmarks (e.g., MedQA-USMLE, PubMedQA), they often fall short in real-world medical scenarios. Specifically, these models frequently struggle to perceive crucial contextual information—such as user identity, key decision-making data, and risk factors—limiting their practical applicability. To address this, we propose the **Multifaceted Self-Refinement** method. This approach leverages the LLM’s self-reflection capabilities to enhance its ability to perceive and actively seek critical contextual information in medical conversations. Notably, our method does not aim to supplement the model with additional medical knowledge; rather, it focuses on improving the model’s capacity to integrate existing encoded medical knowledge with essential contextual cues to solve problems effectively.
>
> Selecting appropriate benchmarks to evaluate this capability is therefore critical. Most existing medical benchmarks are derived from examination-style questions, often in multiple-choice format, which primarily test knowledge mastery. These tasks typically: (1) provide all decision-making information upfront; (2) do not account for the identity of dialogue participants; and (3) require only selecting the most reasonable option without considering additional risk factors. As a result, they fail to capture the model’s awareness of critical medical context, such as asking follow-up questions for missing diagnostic clues or providing personalized recommendations based on user identity.
>
> In contrast, **HealthBench** evaluates LLMs in realistic medical assistant scenarios. Its samples consist of healthcare conversations involving diverse user types (physicians, patients, family members) and various requests (e.g., seeking medical advice or consulting guidelines), effectively testing the model’s medical context-awareness. HealthBench is also highly authoritative: it is annotated by **262 physicians across 60 countries** and widely adopted by multiple LLM research teams (e.g., OpenAI, Kimi, Baichuan) as a benchmark of medical capability. Therefore, it is well suited to validate the effectiveness of our approach.
>
> We also recognize the concern about potential bias from relying on a single benchmark. To address this, during the rebuttal period, we conducted an additional evaluation using **CraftMD** [1]. CraftMD adapts 2,004 questions from existing medical exams into a multi-agent framework, assessing the model’s ability to iteratively inquire about crucial missing context based on limited initial patient input and ultimately complete the diagnosis. Due to time constraints, this supplementary evaluation was conducted on two backbone models (Qwen-32B and Qwen-14B). The results are presented below:
>
> | Method                            | Qwen3-14B | Qwen3-32B |
> | --------------------------------- | --------- | --------- |
> | Original Model                    | 38.2      | 37.3      |
> | +QueryKD                          | 37.3      | 42.8      |
> | **+QueryKD+MultifacetSR (MuSeR)** | **42.5**  | **48.3**  |
>
> The experimental results suggest that our proposed method also boosts the model's accuracy on this evaluation set by **5-6%**. This performance gain illustrates that our approach effectively enhances the model's ability to perceive and proactively query for missing medical decision-making context, thereby substantially improving its performance on the doctor-patient dialogue scenario.
>
> We sincerely appreciate your suggestion once more. We will incorporate these additional experimental results into the revised version of the paper to further support our claims.
>
> [1] Johri et al. An evaluation framework for clinical use of large language models in patient interaction tasks. Nature medicine, 2025.

---

> > ### Author Response · Authors · 2025-12-03
> >
> > 2. **Reliance of GPT-4.1 in evaluation**: Thank you for your careful reading. In fact, HealthBench does not adopt the traditional “LLM-as-a-judge” evaluation approach. Instead, it employs a rubric-based evaluation: each sample in HealthBench is associated with multiple criteria written by human medical experts, which are used to assess the quality of LLM responses. During GPT-4.1 evaluation, GPT-4.1 determines whether each criterion is satisfied based on the LLM’s response, and the final score for each sample is calculated by summing the scores of the satisfied criteria. Compared with the traditional LLM-as-a-judge method, this rubric-based approach offers stronger robustness and consistency. According to the HealthBench paper [2], the authors also examined the performance of GPT-4.1 for evaluation and compared it with human expert assessments. Results shown in Table 5 and Figure 12 of [2] indicate that GPT-4.1 achieves performance comparable to human experts when used for this rubric-based evaluation. Additionally, multiple evaluations were conducted across different models, and Table 7 of [2] reports a standard deviation of approximately 0.002, demonstrating extremely low variability and confirming the reliability of this evaluation approach. We appreciate your comments and will further expand the description of HealthBench in the paper to clarify this point.
> > 3. **"Self-refinement" terminology seems confusing**: Thank you for your thoughtful comments. We refer to our method as “Multifaceted Self-Refinement Learning” primarily because, in principle, it can enhance a model’s medical context-awareness without relying on knowledge distillation. In fact, in our preliminary study, we applied multifaceted self-refinement learning directly to Qwen3-32B and observed approximately a 3% improvement in performance. This gives us reason to believe that applying multifaceted self-refinement learning directly to GPT-OSS-120B could also effectively enhance its medical context-awareness. However, due to computational limitations, we are currently able to perform supervised fine-tuning (SFT) only on models of roughly 30B parameters. In addition, our experiments indicate that distilling knowledge from a stronger teacher model into smaller models can effectively inject medical knowledge, which further amplifies the benefits of self-refinement learning. We will further clarify this point in our revised paper.
> > 4. **Reference for lines 42-43**: Thank you. The claim you mentioned refers to the sentence in our paper: *“Despite these advancements, LLMs still struggle to meet the demands of real-world medical applications, limiting their practical utility in healthcare settings.”* In fact, recent medical benchmarks [1, 3–4] have attempted to evaluate existing large models using tasks that more closely resemble real clinical scenarios (e.g., consultation tasks based on limited initial information). These studies found that, although the models perform well on QA-based benchmarks, they struggle in complex, scenario-based tasks, which limits their applicability in real clinical settings. We will incorporate these references to support this claim in our revised paper.
> > 5. **Code Repo Issue**: Thank you for your comments. In fact, we belong to the R&D team of a commercial company. Immediately after submitting the paper, we requested approval to open-source the code inside our company. However, as of the start of the review process, the approval had not yet been granted, so we were unable to submit the code to a public repository at that time. We have now obtained the approval and have made the code publicly available in a repository. We also plan to release the dataset and model weights alongside the code after the paper is finalized.
> >
> > [2] Arora et al. HealthBench: Evaluating Large Language Models Towards Improved Human Health.
> >
> > [3] Hager et al. Evaluation and mitigation of the limitations of large language models in clinical decision-making. Nature Medicine, 2024.
> >
> > [4] Zhou et al. Evaluating LLMs Across Multi-Cognitive Levels: From Medical Knowledge Mastery to Scenario-Based Problem Solving. ICML 2025.

---

> > > ### Author Response · Authors · 2025-12-03
> > >
> > > 6. **Issue of "the formulations above represent our design goals rather than explicit optimization objectives"**:  Thank you for your careful reading. In Section 3.1, we formulate the objectives of the three core modules of our method—query generator, multifaceted self-refinement, and SFT learning. These three objectives are derived from the main objective in Equation 1 and represent high-level objectives for improving the model’s medical capabilities through data construction. However, since the true query distribution and the ideal response distribution are both inaccessible, Equations 2 and 3 cannot be directly optimized. Therefore, in our implementation, we approximate these optimization goals through the attribute-conditioned query generator and multifaceted self-refinement. Although these two objectives cannot be directly used for optimization, we believe they provide methodological guidance for the design of our approach and offer valuable inspiration for future work in enhancing model medical capabilities via data synthesis. Therefore, we keep these objectives in the problem formulation section.
> > > 7. **Theoretical insights for "student models can outperform the teacher model"**: Thank you for your insightful comments. This phenomenon is indeed very interesting. We believe it arises because the factors influencing a model’s medical capability can be divided into **medical knowledge coverage** and **ability to apply medical knowledge**. During the knowledge distillation stage, the medical knowledge and reasoning abilities of the teacher model are distilled into the student models, allowing the student models’ knowledge coverage to approach that of the teacher. In the multifaceted self-refinement stage, the student models’ medical context-awareness is further enhanced, improving their ability to apply medical knowledge to solve real-world problems. As a result, even though the student models may not match the teacher in terms of knowledge coverage, their ability to apply medical knowledge surpasses that of the teacher, leading to overall higher performance on HealthBench. We appreciate your comments and will discuss this phenomenon in more detail in the revised paper.

---

### Official Review · Reviewer_mzCw · 2025-11-01

**Soundness:** 3
**Presentation:** 3
**Contribution:** 3
**Rating:** 4
**Confidence:** 5

**Summary:**

This paper aims to enhance the Medical Context-Awareness Ability of LLMs via Multifaceted Evaluation Metrics to refine the generation so that the model can learn to capture three key facets (decision-making, communication, and safety) on HealthBench. By this data-driven pipeline, they successfully improved the healthcare response generation ability of smaller LLMs and even surpassed the biggest competitor through continual distillation from gpt-oss even on the hard subset.

**Strengths:**

1. This paper focuses on a data-driven pipeline to use multifaceted eval metrics to refine the generation. This process ensures the diversity and quality of the data used for further reinforcement training.

2. Their results show significant improvements of their pipeline on improving the health capabilities of smaller LLMs, making the data potentially useful for this domain.

3. Health is a domain that requires more careful eval, and this paper focuses on decision-making, communication, and safety, which are essential to enhance the safety and usefulness of health questions.

**Weaknesses:**

1. Why only use GPT-oss-120B as the teacher? how about using other models? GPT-oss-120B might not be the best

2.  Why choose those three key facets (decision-making, communication, and safety) but not other dimensions? I want to learn more scientific justification for this choice.

**Questions:**

1. Why use DeepSeek-V3 for generating the queries but use smaller models ((Qwen3-14B/32B or OpenPangu-7B) as the multifaceted selfrefinement module? The inference cost is minimal, and I did not see any specific reason for using smaller models here. especially OpenPangu-7B is not that good.

2. The authors mentioned "capture the diversity and complexity of real-world medical queries". How do you measure diversity? quantitively?

3. What are the obvious failure cases after this pipeline? it would be good to provide a more thorough analysis to help us understand the limitations of this pipeline.

---

> ### Author Response · Authors · 2025-12-03
> **Response to Reviewer mzCw**
>
> We sincerely appreciate your thorough review. Below, we address each of your concerns in detail.
>
> 1. **Teacher model Issue**: Thank you very much for your comments. Indeed, some LLMs (e.g., o3, GPT-5-thinking) exhibit stronger capabilities in medical scenarios compared to GPT-OSS-120B. In theory, using them as teacher models could further enhance the medical abilities of the student model and further improve the effectiveness of our proposed multifaceted self-refinement approach. However, from a practical perspective, these stronger LLMs are mostly **proprietary**. Although they can be accessed, they only return the final response and do not expose their intermediate reasoning chain to the user. As a result, they are difficult to use for distilling knowledge and reasoning capabilities into student models.
>
>    On the other hand, we have tried many open-source models and found that GPT-OSS-120B demonstrates the strongest medical capability, which is why we selected it as the teacher model to inject knowledge into the student model. We also conducted preliminary experiments using other open-source reasoning models (e.g., DeepSeek-R1) as teacher models; the results are shown in the table below:
>
>    | Method                            | GPT-OSS-120B as teacher model | DeepSeek-R1 as teacher model |
>    | --------------------------------- | ----------------------------- | ---------------------------- |
>    | Backbone (Qwen3-14B)              | 43.9                          | 43.9                         |
>    | +QueryKD                          | 55.9                          | 51.2                         |
>    | **+QueryKD+MultifacetSR (MuSeR)** | **61.8**                      | **53.0**                     |
>
>    The experimental results show that, although using DeepSeek-R1 as the teacher model can also effectively improve the student model’s performance and our proposed method can further enhance it, the magnitude of improvement is smaller than when GPT-OSS-120B is used as the teacher. We speculate that this may be because GPT-OSS-120B can generate longer chains of thought, resulting in more comprehensive and well-formed medical responses. Thank you again for your comments. We will incorporate this additional experiment in our revised paper and further clarify our choice of the teacher model.

---

> > ### Author Response · Authors · 2025-12-03
> >
> > 2. **Justification for the chosen facets**: Thank you for your thoughtful comments. The primary motivation of this work is that although existing LLMs generally achieve strong performance on medical benchmarks, they still perform unsatisfactorily in real-world medical scenarios (e.g., medical conversations), particularly due to the deficiencies in perceiving key medical contextual information. In this paper, we focus on key facets of medical context-awareness by referring the evaluation axes of the authoritative HealthBench [1], and also consider the shortcomings in medical capabilities exposed by several medical evaluation studies. Based on these insights, we select **decision-making**, **communication**, and **safety** as three key medical context-awareness facets:
> >
> >    **Decision-making**: Decision-related contextual information (e.g., patient demographics, symptoms, medications, medical history) is essential for a medical assistant to provide helpful responses. Existing medical benchmarks typically provide all necessary decision-making information within the prompt, and LLMs tend to perform well under such conditions. However, in real-world settings (e.g., patient consultations), users rarely provide complete decision-related information at the outset (either due to missing diagnostic tests or simply not knowing what information is relevant for further decision-making). However, current LLMs often struggle to provide helpful responses when initial decision-making information is insufficient [2–3]. Therefore, we include decision-making as one dimension of context-awareness, aiming to enhance the model’s ability to proactively request missing information under incomplete initial contexts.
> >
> >    **Communication**: Communication-related contextual information (e.g., user identity, geographic location) is also crucial for generating helpful medical responses. This includes perceiving the user’s identity (e.g., patient vs. clinician) to adjust the level of medical terminology and detecting location information to reason about local resource availability (e.g., certain diagnostic tests may not be accessible in specific regions). Effective perception of such contextual cues is fundamental for providing personalized medical advice.
> >
> >    **Safety**: The medical domain adheres to the strict principle of non-maleficence, meaning that providing safe medical guidance is a core requirement for any medical assistant. To ensure safety, a model must be able to identify potential risk factors in user inputs and alert users to possible hazards. However, prior work [4] shows that existing LLMs often lack an understanding of the relative weight and priority of risk factors, which leads to difficulties in handling complex and uncertain real-world clinical scenarios. Therefore, we consider the effective perception of risk factors a key dimension of medical context-awareness as well.
> >
> > 3. **"Why use DeepSeek-V3 for generating the queries but use smaller models as the multifaceted self-refinement module"**: Thank you very much for your thoughtful comments and careful reading. It may indeed seem counterintuitive to use smaller models for multifaceted self-refinement. However, in our implementation, the “smaller models” are actually distilled models that are finetuned on medical queries generated by DeepSeek-V3 and responses generated by the teacher model (GPT-OSS-120B). These distilled models have already achieved performance on HealthBench that is comparable to or even surpasses DeepSeek-R1 (even the lowest-performing OpenPangu-7B achieves a score of 53.0 after distillation, compared to 53.9 for DeepSeek-R1). Therefore, using these distilled smaller models in the multifaceted self-refinement module actually yields better results than using DeepSeek-V3 directly. Moreover, considering that the second stage involves multiple inferences (reflection across the three facets and answer revision) and the inputs are significantly longer, the computational cost is significantly higher than that of query generation. Using smaller models in this module can also substantially reduces the computational cost. Thank you for your comments, and we will clarify this point further in the revised paper.
> >    Once again, we appreciate your suggestions. We will incorporate the above justification into the revised paper to support our selection of context-awareness facets.
> >
> > [1] Arora et al. HealthBench: Evaluating Large Language Models Towards Improved Human Health.
> >
> > [2] Hager et al. Evaluation and mitigation of the limitations of large language models in clinical decision-making. Nature Medicine, 2024.
> >
> > [3] Zhou et al. Evaluating LLMs Across Multi-Cognitive Levels: From Medical Knowledge Mastery to Scenario-Based Problem Solving. ICML 2025.
> >
> > [4]Christof et al. Implications of integrating large language models into clinical decision making. Communications Medicine, 2025.

---

> ### Author Response · Authors · 2025-12-03
>
> 4. **Clarification of "capture the diversity and complexity of real-world medical queries"**: Thank you for your comments. In the paper, our original wording is: *“These attributes are chosen to capture the diversity and complexity of real-world medical queries.”* What we intended to convey here is the design goal of our query generator: by sampling key attributes, we aim to approximate the diversity and complexity of real-world medical queries. Indeed, we agree that real-world medical queries are inevitably more diverse and complex than the queries we synthesize. Nevertheless, our experiments show that performing knowledge distillation directly on queries generated using our method allows the student models to approach the performance of the teacher model on HealthBench (e.g., 56.6 for Qwen3-32B after distillation vs. 57.6 for its teacher model GPT-OSS-120B). This suggests that the medical queries synthesized by our method already exhibit a substantial degree of diversity. Of course, further developing quantitative measurements of query diversity and continuously improving the query generation process could potentially enhance our method’s effectiveness, while we insist that the currently synthesized queries are already sufficient to support our multifaceted self-refinement approach.
>
> 5. **Bad Case Analysis**: Thank you for your valuable suggestion. In our paper, we provide a detailed analysis of our method’s performance across different axes and themes in HealthBench. We found that our approach achieves performance improvements across all themes and on all axes except for the communication quality axis. For the communication quality axis, we further conducted a bad case analysis and found that the majority of errors (>99%) were due to the model producing overly long responses. For example, for the following query:
>
>    > simple writing the patient complains of itchy red rash after starting new medication physical exam welts on arms assessment likely drug eruption plan discontinue medication
>
>    The response provided by Qwen3-32B after trained by the proposed method tend to be very long:
>
>    > **Subjective** – 30‑year‑old male (example) – “I have an itchy, red rash that started ≈ 2 days after I began taking ...
>
>    > **Objective** – General appearance: comfortable, vitals within normal limits ...
>
>    > **Assessment** – **Probable drug eruption (exanthematous drug‑induced rash).**...
>
>    > **Plan**
>    >  1. **Discontinue **[drug name]** immediately.**  ...
>    >  2. **Symptomatic relief** ...
>    >  3. **Monitoring & Follow‑up** ...
>    >  4. **Safety instructions**  ...
>    >  5. **Documentation** – Notify the prescribing provider of the reaction ...
>
>    > **Disclaimer** ...
>
>    Through analysis, we found that these overly long responses are due to the teacher model (GPT-OSS-120B) tending to provide detailed answers to user queries through lengthy explanations and tables, a characteristic inherited by the distilled model. While longer responses can enhance completeness to some extent, they may also result in unnecessary verbosity, which can negatively affect the user’s reading experience. We appreciate your feedback and will include this bad case analysis in our revised paper. In future work, we plan to further optimize the reflection prompts for the communication facet and incorporate RL training with length-based rewards to improve the model’s ability to dynamically adjust response length according to user needs.

---

### Official Review · Reviewer_HNZe · 2025-11-02

**Soundness:** 3
**Presentation:** 2
**Contribution:** 2
**Rating:** 4
**Confidence:** 4

**Summary:**

The paper introduces MuSeR, a data-driven framework to improve medical context-awareness in LLMs by synthesizing diverse, attribute-conditioned queries (e.g., user role, region, intent, information completeness) and training models to self-evaluate and refine their answers along three facets: decision-making, communication, and safety. For each synthetic query, the model produces an initial reply, generates facet-specific rationales, and directly refines the answer based on those rationales; the refined pairs then supervise SFT. A query-guided knowledge distillation stage precedes SFT to transfer medical knowledge from a stronger teacher, boosting downstream effectiveness. On HealthBench, MuSeR substantially improves multiple open-source backbones and achieves new open-source SOTA, with especially large gains on the context-awareness axis.

**Strengths:**

The results are strong and well-documented. MuSeR delivers consistent, sizable gains on HealthBench across multiple backbones, with Qwen3-32B and Qwen3-14B improved to 63.8% and 61.8%, surpassing a stronger teacher and setting a new open-source SOTA; detailed plots also show improvements on the hard subset and across evaluation axes/themes, emphasizing context-awareness. The paper backs these outcomes with granular analyses: stage-wise and multi-faceted self-refinement and facet ablations. Overall, the combination of broad backbone coverage, competitive headline scores, and thorough ablations/case study makes the empirical case both convincing and reproducible.

**Weaknesses:**

Weaknesses*: Backbone coverage is narrow; most results center on Qwen, so it’s unclear whether gains transfer to other families. Adding full runs on Llama-3/Mistral/Qwen2.5 (multiple sizes) would clarify generality. On novelty and positioning, the attribute-conditioned query synthesis overlaps with prior medically conditioned instruction tuning (e.g., AlpaCare’s diverse, synthetic medical instructions [1]); this prior work should be cited and contrasted to specify what is new here beyond the chosen attribute schema and prompts. Likewise, the use of rationale/CoT distillation follows earlier explanation-distillation work [2] and should be discussed. Finally, the empirical scope is largely HealthBench; adding at least one more public medical benchmark (e.g., MedQA/USMLE or PubMedQA) would help validate robustness outside HealthBench’s distribution.

References:
[1] Zhang et al. AlpaCare: Instruction-tuned Large Language Models for Medical Application.
[2] Li et al. Explanations from Large Language Models Make Small Reasoners Better.

**Questions:**

The current self-refinement loop is trained with SFT. Have you explored an RL variant (e.g., PPO/GRPO) where facet-specific rewards (decision-making, communication, safety) are used directly as the objective? In addition, for training data, any decontamination test of Healthbench is done?

---

> ### Author Response · Authors · 2025-12-03
> **Response to Reviewer HNZe**
>
> We sincerely appreciate your thorough review. Below, we address each of your concerns in detail.
>
> 1. **Backbone Issue**: Thank you very much for your suggestions. Our work mainly proposes a multifaceted self-refinement approach, which constructs training data by guiding the target LLM to self-reflect from multiple dimensions (decision-making, communication, and safety). In doing so, the model’s medical context awareness and scenario-based problem-solving ability can be effectively enhanced through SFT on the refined data. Since our method is designed primarily for reasoning LLMs, and Qwen3 is currently one of the most popular open-source reasoning LLMs with strong performance across multiple domains, we selected Qwen3 models of different sizes (14B / 32B) for experimentation. Meanwhile, we also noticed that using a single backbone may not be sufficient to demonstrate the generalizability of our approach. Therefore, we further included a model from another family (**OpenPangu-7B**) in our experiments. The results show that our method can significantly improve the medical context awareness capability of models with different backbones.
>
> Based on your comments, we further validated the effectiveness of our method on another two backbones: R1-distilled-Llama-8B and R1-distilled-Qwen-14B. The reason we selected these two models is that, on one hand, they were distilled on DeepSeek-R1 data and thus possess reasoning patterns that fit well with our method; on the other hand, they are trained on different base models (**Llama3-8B and Qwen2.5-14B**), which allows us to further verify the generalizability of our approach. The experimental results are shown in the table below:
>
> | Method                            | R1-distilled-Llama-8B | R1-distilled-Qwen-14B |
> | --------------------------------- | --------------------- | --------------------- |
> | Original Model                    | 14.1                  | 25.5                  |
> | +QueryKD                          | 54.4                  | 56.8                  |
> | **+QueryKD+MultifacetSR (MuSeR)** | **56.3**              | **60.7**              |
>
> These results further demonstrate that our method can be applied to models from multiple families to enhance their medical context awareness. Thank you again for your valuable feedback. We will include the corresponding additional results in the revised version of the paper.

---

> > ### Author Response · Authors · 2025-12-03
> >
> > 2. **Comparison with Prior Works**: We sincerely appreciate your thoughtful comments. The two papers you mentioned are indeed related to certain components of our method, but they differ significantly from us in both motivation and implementation details:
> >
> > **Medical Instruction Synthesis**: Existing methods for synthesizing medical instructions, such as AlpaCare [1], primarily followed the Self-refine method [2]: These methods utilize a fixed data synthesis prompt and a few sampled clinician-curated seed tasks to generate a variety of medical instructions using a powerful LLM (e.g., GPT-4). For example, the prompt used in AlpaCare  is provided below:
> >
> > > Your objective is to generate diverse medical-related tasks.
> > >
> > > Here are the requirements:
> > >
> > > 1.Ensure that all tasks are related to the medical domain.
> > >
> > > 2.Craft tasks that encompass varied points of view, e.g.
> > >
> > > experts, students and patients, etc.
> > >
> > > ...
> > >
> > > 15.Each task should adhere to the following structure:
> > >
> > > ’Type: \n, Topic: \n, View: \n, Difficulty: \n, Instruction:
> > >
> > > \n, Input: ’. Start each new task with ’###’.
> > >
> > > List of 15 tasks:
> > >
> > > [Seed task 1]
> > >
> > > [Seed Task 2]
> > >
> > > [Seed Task 3]
> >
> > While these methods are effective for enhancing LLMs' instruction-following ability in the medical domain, the distribution of critical attributes (such as user identity and information ambiguity) in the generated queries is implicitly affected by the seed tasks and the inherent biases of the generator LLM, which may potentially limits the diversity of generated queries over these key attributes.
> >
> >  In this paper, we focus on medical conversational scenarios and design an attribute-conditioned query generator that produces queries based on sampled key attributes (e.g., dialogue topic, user intent, user identity) instead of seed tasks. We use the following prompt to generate queries based on sampled attributes, which can be found in Figure 9 of our paper:
> >
> > > Assume that you are a user that seeks for help from a medical large language model. Please generate a query to the LLM, following the constraints and instructions below.
> > >
> > > Constraints:
> > >
> > > \- User role: <<<role>>>
> > >
> > > \- User region: <<<region>>>
> > >
> > > \- Disease or injury: Your query should be related to "<<<disease>>>"
> > >
> > > \- User goal (intention): <<<intention>>>
> > >
> > > \- Vagueness level of the query: <<<vagueness>>>
> > >
> > > \- Completeness level of the query: <<<completeness>>>
> > >
> > > \- Language style: <<<language_style>>>
> > >
> > > Instructions:
> > >
> > > 1. Make sure the query is directly relevant to the given disease or injury stated goal.
> > > 2. The query should be consistent with all the specified constraints above.
> > > 3. The output should be a **single natural query** a real user would input to an LLM, not a
> > >
> > > list or explanation. Directly output the query without any additional commentary or notes.
> >
> > Such an attribute-conditioned query generator allows us to explicitly control the query distribution through the sampling process, ensuring sufficient diversity of queries in the medical conversation scenario.
> >
> > **Rationale/CoT distillation**: the rationale distillation methods [3] you kindly mentioned aim to leverage LLMs to generate explicit CoT/rationales based on existing question-answer pairs. This generated textual knowledge is then used to train small models via a multi-task learning  framework to improve their reasoning performance. The typical process involves: (1) The teacher model generates high-quality CoT explanations for the training data using various techniques (COTE, RP, CROP mentioned in [3]); (2) The student model is then jointly fine-tuned using a multi-task learning framework, compelling it to simultaneously learn to generate the reasoning process (rationale) and predict the final answer.
> >
> > In contrast, our proposed *query-based knowledge distillation* adheres to the conventional *response-based distillation* paradigm. We utilize synthetic queries as the distillation data and directly prompt the teacher model to generate complete responses. The crucial distinction from traditional response-based methods lies in the selection of the teacher model: we employ a reasoning LLM (trained to automatically produce its rationale prior to the final answer). As a result, the teacher's output inherently contains both the model’s rationale and the final prediction, naturally satisfying the requirements for knowledge transfer without necessitating a separate, explicit rationale generation step.
> >
> > Thank you again for your comments. We will incorporate a discussion of these prior works in the Related Work section of the revised paper.
> >
> > [1] Zhang et al. AlpaCare: Instruction-tuned Large Language Models for Medical Application. SCI-FM 2025.
> >
> > [2] Wang et al. Self-instruct: Aligning language models with self-generated instructions. ACL 2023.
> >
> > [3] Li et al. Explanations from Large Language Models Make Small Reasoners Better. SAI-AAAI2024.

---

> > > ### Author Response · Authors · 2025-12-03
> > >
> > > 3. **Other Benchmark Issue**: We are profoundly grateful for your insightful suggestions.
> > >
> > > The motivation for our work stems from the finding that while existing LLMs achieve considerable performance on established medical benchmarks (such as MedQA-USMLE, PubMedQA, etc.), when we apply these models in real-world medical business scenarios, they often lack the essential ability to perceive critical contextual information (e.g., user identity, crucial decision-making data, risk factors), thus failing to meet practical application demands.
> > >
> > > Therefore, we propose the Multifaceted Self-refinement method. This approach leverages the LLM's self-reflection mechanism to enhance its capacity for perceiving and actively seeking critical contextual information within medical conversation settings. It is important to note that our method is not designed to supplement the model with additional medical knowledge; rather, it aims to boost the model's ability to effectively integrate encoded medical knowledge with key medical context to solve problems.
> > >
> > > To validate the effectiveness of our proposed method, it is crucial to select appropriate benchmarks that can effectively assess the model's medical contextual awareness. The vast majority of existing medical benchmarks are constructed primarily from medical examination questions, often in multiple-choice format, which largely assess the model's mastery of medical knowledge. However, these tasks typically: (1) provide all decision-making information upfront; (2) do not involve the identity of the dialogue participants; and (3) only require selecting the most reasonable option without considering additional risk factors. Consequently, they fail to reflect the model's awareness to critical medical context, such as the ability to ask follow-up questions for missing diagnostic clues or to provide personalized recommendations based on user identity.
> > >
> > > In contrast, **HealthBench** focuses on evaluating LLMs' performance as medical assistants in real-world scenarios. Its evaluation samples are in the form of healthcare conversations, involving diverse user types (physicians, patients, family members, etc.) and various types of requests (seeking medical advice, consulting guidelines, etc.), which is able to reflect the model's medical context-awareness. Furthermore, HealthBench stands as one of the most authoritative medical leaderboards, as it is annotated by **262 physicians across 60 countries** and is widely adopted by multiple LLM research teams (e.g., OpenAI, Kimi, Baichuan) as a benchmark of medical capabilities. Therefore, HealthBench is highly suitable for verifying the effectiveness of our approach.
> > >
> > > We also fully understand your concern regarding the potential bias introduced by relying on a single evaluation benchmark. Thus, during the rebuttal period, we further evaluated our method on another medical benchmark, **CraftMD** [4]. The CraftMD framework adapts 2,004 questions from existing medical exam benchmarks, using a multi-agent setting to assess the model's capacity to continuously inquire about crucial missing context based on a patient's limited initial input in the doctor-patient dialogue scenario, ultimately completing the diagnosis. Due to time constraints, we conducted this supplementary evaluation on two backbone models (Qwen-32B/14B) using this benchmark. The results are presented below:
> > >
> > > | Method                            | Qwen3-14B | Qwen3-32B |
> > > | --------------------------------- | --------- | --------- |
> > > | Original Model                    | 38.2      | 37.3      |
> > > | +QueryKD                          | 37.3      | 42.8      |
> > > | **+QueryKD+MultifacetSR (MuSeR)** | **42.5**  | **48.3**  |
> > >
> > > The experimental results demonstrate that our proposed Multifaceted Self-refinement (MuSeR) method further boosts the model's accuracy on this evaluation set by **5-6%**. This performance gain illustrates that our approach effectively enhances the model's ability to perceive and proactively query for missing medical decision-making context, thereby substantially improving its performance on the doctor-patient dialogue scenario.
> > >
> > > We sincerely appreciate your suggestion once more. We will incorporate these additional experimental results into the revised version of the paper to further substantiate our claims.
> > >
> > > [4] Johri et al. An evaluation framework for clinical use of large language models in patient interaction tasks. Nature medicine, 2025.

---

> > > > ### Author Response · Authors · 2025-12-03
> > > >
> > > > 4. **Exploration on RL**: Thank you very much for your valuable feedback. In this work, we primarily chose to use SFT rather than RL for multifaceted self-refinement since SFT is substantially more computationally efficient. Our multifaceted self-refinement approach requires the LLM to reflect on and improve the initial response from multiple perspectives. If implemented with RL, this would require generating multiple facet-specific rewards (e.g., decision-making, communication, safety) through LLM-as-a-judge. For each training batch, this process would require **batch_size × rollout_size × num_facets** inference calls, which leads to significantly higher computational overhead compared to rule-based RL or standard SFT. Furthermore, we found in our experiments that SFT on self-refined training data is already sufficient to substantially improve the model’s medical context-awareness capability.
> > > >
> > > >    After the paper submission, we also explore combining RL with the proposed multifaceted self-refinement method. Specifically, we conducted a preliminary study by selecting Qwen3-14B as the backbone model and modifying the prompts used in our method to generate facet-specific rewards. We then used Qwen3-32B to produce these rewards and performed a small-scale RL training on 10k synthetic queries. The experimental results are shown below:
> > > >
> > > >    | Method               | Qwen3-14B |
> > > >    | -------------------- | --------- |
> > > >    | Backbone (Qwen3-14B) | 43.9      |
> > > >    | +MultifacetedSR (RL) | **46.5**  |
> > > >
> > > >    The experimental results show that integrating RL with our proposed multifaceted self-refinement framework can also effectively enhance the backbone model’s capabilities in medical conversational settings (+2.6%). We sincerely appreciate your comments, and we will further explore incorporating RL into our method in future work to further improve the model’s medical context-awareness capability.
> > > >
> > > >
> > > >
> > > > 5. **Decontamination Test Issue**: We sincerely appreciate your constructive suggestions. Data contamination can indeed affect the fairness of evaluation; however, our work is not impacted by this issue for the following reasons:
> > > >
> > > >    1. HealthBench contains only queries and their corresponding rubrics, but no golden labels (i.e., ideal responses to the queries). Therefore, even if existing LLMs were trained on HealthBench, they would not be able to improve performance simply by memorizing answers.
> > > >    2. HealthBench is a very recent dataset, released on May 12, 2025, which makes it unlikely that existing LLMs have been trained on it.
> > > >    3. Based on your suggestions, we further conducted a decontamination test. Specifically, for each conversation in HealthBench, we computed its N-gram Jaccard similarity with each of our synthetic queries and took the maximum similarity as the overlap score for that conversation. We then averaged the overlap scores over all 5,000 HealthBench conversations to quantify the repetition ratio between HealthBench and our synthetic queries. For comparison, we also performed the same calculation on Alpaca[2], a widely used general-domain SFT dataset. The results are as follows:
> > > >
> > > > | N-gram | Our Synthetic Queries | Alpaca |
> > > > | ------ | --------------------- | ------ |
> > > > | N=2    | 5.3%                  | 4.0%   |
> > > > | N=3    | 3.0%                  | 1.8%   |
> > > > | N=4    | 1.7%                  | 0.8%   |
> > > >
> > > > We found that our synthetic queries exhibit low overlap scores with HealthBench queries across different N-gram settings, and are only slightly higher than the overlap scores with Alpaca (which may be because Alpaca is a general-domain dataset, whereas our queries are in the medical domain). This indicates that HealthBench is not likely leaked into our training dataset. We will include this decontamination test in the appendix of our revised paper.

---

### Meta-Review · Area_Chair_b3zC · 2026-01-05

**Summary:**

The reviewers all felt that the paper tackles an important problem with a practically useful pipeline. But the concerns lay in narrow evaluation, limited generality and under-justified design choices.

The paper has strong real world medical relevance and the data pipeline appears to be principled. The weaknesses include relying only on HealthBench, and using LLM as a judge. It is unclear for the reviewers if the gains are architecture-specific and some justifications are not clearly specified -- including error taxonomies, clinician interviews etc. Ablation studies can significantly strengthen the paper. Some of the reviewers percieved the contributions as incremental.

**Reviewer Concerns:**

The authors have taken significant effort in addressing the concerns in the rebuttal. For any other conference, the space limitations would ahve restricted the response but it does appear like the original paper missed a lot of important details.

**Reviewer Scores:**

None of the reviewers responded (even before the leak issue). However, I want to think that they still would not have championed the paper and so the paper would still only be on a borderline.

---

### Decision · Program_Chairs · 2026-01-26

Reject